# Adapting to the Impacts Posed by Climate Change: Applying the Climate Change Risk Indicator (CCRI) Framework in a Multi-Modal Transport System

Tianni Wang [1], Mark Ching-Pong Poo [2,*], Adolf K. Y. Ng [3] and Zaili Yang [2]

1 College of Transport and Communications, Shanghai Maritime University, Shanghai 201308, China; wangtn@shmtu.edu.cn
2 Liverpool Logistics, Offshore and Marine Research Institute, Liverpool John Moores University, Liverpool L3 5UX, UK; z.yang@ljmu.ac.uk
3 Faculty of Business and Management, BNU-HKBU United International College, Zhuhai 519088, China; adolfng@uic.edu.cn
* Correspondence: c.p.poo@ljmu.ac.uk

**Abstract:** Climate change has threatened the infrastructure, operation, policymaking, and other pivotal aspects of transport systems with the accelerating pace of extreme weather events. While a considerable amount of research and best practices have been conducted for transport adaptation to climate change impacts, there is still a wide gap in the systematic assessment of climate risks on all-round transport modes (i.e., road, rail, sea, and air) with a comprehensive review and a quantitative scientific framework. This study aimed to critically review studies on how the transport sector has adapted to the impacts posed by climate change since the dawn of the 21st century. To support climate risk assessment in comprehensive transport systems, we developed a Climate Change Risk Indicator (CCRI) framework and applied it to the case of the British transport network. Focusing on a multi-modal transport system, this offers researchers and practitioners an invaluable overview of climate adaptation research with the latest tendency and empirical insights. Meanwhile, the developed CCRI framework elaborates a referable tool that enables decision-makers to employ objective data to realise quantitative risk analysis for rational transport adaptation planning.

**Keywords:** climate adaptation; transport infrastructure; risk assessment; multi-modal transport system

## 1. Introduction

Transport is a crucial component for humans and society [1]. However, many factors, including the economy and public health, affect the system by presenting different disruptions. Furthermore, climate change is one of the most threatening issues influencing human activities because it could significantly reduce the efficiency of transport systems [2]. The goals of Agenda 2030, also known as the Sustainable Development Goals (SDGs), include a range of objectives related to climate change adaptation [3]. SDG 11, "Make cities and human settlements inclusive, safe, resilient, and sustainable", is particularly relevant to climate change adaptation for transport infrastructure. This SDG includes the target to "strengthen efforts to protect and safeguard the world's cultural and natural heritage", which can be achieved through measures such as improving the resilience of transport infrastructure to climate change impacts. Furthermore, SDG 13, which aims to "take urgent action to combat climate change and its impacts", is relevant to climate change adaptation for transport infrastructure, as discussed earlier. This SDG aims to "integrate climate change measures into national policies, strategies, and planning", which can help mainstream climate change adaptation considerations in terms of transport infrastructure planning and design.

Therefore, many scholars, such as Becker [4–11] and Schweikert [12–15], have tended to investigate the relationships between transport and climate change. The focus has been

two-fold: mitigation in domination and adaptation with a growing profile. In terms of mitigation, new energy alternatives, such as electric vehicles [16,17] and ships [18,19], and tactical management, such as speed control [20] and reverse logistics [21], have been used to promote decarbonisation with diverse supports, including the application of urban policies [22] and the circular economy concept [23].

In the post-pandemic era, the stresses between transport and climate change are tightening due to complex natural and human factors which increase the vulnerabilities of the urban system [24]. As far as adaptation is concerned, the analysis of climate threats and risks is the first and foremost step. Different transport modes face various climate threats to different extents [25]. A threat could affect several areas and cause damage and transport disruptions with enormous economic losses. For instance, a significant drop in water level occurred in the Port of Montreal, QC, Canada, and impacted its transport networks, including requiring a reduced tonnage per trip, resulting in an increased number of trips and traffic backup [26]. A further decrease in the water level by 0.5–1.0 m was expected to result in an economic loss of over USD1.9 billion by 2050. In the USA, research showed that the temperature increased by 1.5 degrees Fahrenheit annually from the year 2014 to 2015 [27]. Drivers, pedestrians, and bikers who are more likely to go out in warmer weather accounted for over 20% of the increase in road deaths in 2015 [28].

In the UK, there was a 70% increase in flooding events from 1998 to 2009 [29], and a significantly wetter period occurred, prolonging the flooding caused by intensive rainfall from 2013 to 2016. A catastrophic flood that occurred in Cumbria in 2015 broke the precipitation records from 2009 with 341.4 mm of rainfall [30]. Roads were shut in the severely affected areas, and over 100 bridges were damaged or destroyed. In October 2017, the floods between Carlisle and Maryport resulted in enormous disruptions and the blockage of rail lines, estimated to have caused over US$1.3 billion in damages, and claimed 18 lives [31]. In addition, the temperature at London Heathrow Airport in the summer of 2020 reached 37.8 °C, which was recorded as the UK's third hottest day in history. The Meteorological Office UK confirmed that the August 2020 heatwave broke temperature and duration records. One of the most common impacts is the melting of roads which puts heavy pressure on maintenance [32]. Based on the prediction by the UK Hadley Centre for Climate Change Prediction Research that there would be a 4 °C rise in the global temperature by the end of this century, it was expected that temperature-related accidents would cause approximately 600 additional deaths annually, equating to a cost of US$60 billion from 2010 to 2099 [33]. In 2021, wildfires at White Rock Lake disrupted Kelowna International Airport in BC, Canada, where more than 40 flights were cancelled over 24 h [34].

There is no scarcity of research on climate adaptation in the transport sector (e.g., [31,35–38]). Nonetheless, the effect of climate change on transport is multi-dimensional, given the popularity of multi-modal transport in the containerised freight sector. A container shipment often combines sea, road, rail, and possibly air transport segments in an established supply chain for door-to-door service from the manufacturer to the end user. The current studies on transport's adaptation to climate change are lacking integration between different transport modes. Therefore, there is an urgent need to provide an overview of the development of a new holistic climate adaptation framework across different transport modes.

Hence, it is crucial to understand and define the latest developments of related academic studies, which can provide a comparative analysis of the development of different transport modes in the climate adaptation context for cross-sector fertilisation. Understanding such, this study critically reviewed the research on how the transport sector has adapted to the impacts posed by climate change since the dawn of the 21st century. Based on this analysis, we developed a Climate Change Risk Indicator (CCRI) framework to facilitate climate adaptation in a comprehensive transport network. This provides a platform to first compare the climate resilience and vulnerability of different transport modes and realise multi-modal adaptation planning.

The novelty of this work lies in (1) conducting a critical review of studies on how the transport sector adapts to the impacts of climate change across different transport modes and (2) developing a new holistic framework for a comprehensive transport network. It will facilitate climate adaptation by comparing the climate resilience and vulnerability of different transport modes and enabling multi-modal adaptation planning. It provides a comparative analysis of the development of different transport modes in the climate change adaptation context, which can be helpful for cross-sector fertilisation. It also generates practical implications on rational transport adaption resource management for transport authorities/operators including logistics managers of whole container supply chains involving multiple transport modes.

The rest of this paper is structured as follows: A systematic literature review is presented in Section 2 to provide insights for upcoming climate change adaptation studies. In Section 3, the CCRI framework is developed to oversee the possibility of integrating different transport modes for further development. Finally, the new framework is demonstrated in a case study of the British transport systems in Section 4, before presenting the conclusion in Section 5.

## 2. Literature Review

### 2.1. Elaboration of a Corpus on Climate Adaptation Research in the Transport Sector

Different from previous research, this study has filled the gap of current climate adaptation studies focusing only on individual segments of a whole transport chain/network, such as inland transport [39] or seaports [40]. From this perspective, it makes new contributions to the holistic analysis of all the transport segments for both cross-enrichment among different transport modes and integrated adaptation planning from a supply chain perspective. A database was first prepared to collect all the relevant articles on climate change impacts and adaptation research in the transport sector tracing back to 2000, when it was evident that the fast growth of climate adaptation emerged in the transport sector [39].

Taking the "Web of Science" and "Scopus" as the leading academic research platforms, this study was conducted by following three key phases: (1) a systematic literature review [41] to select, screen, and refine the representative papers; (2) publication trend analysis [42] to identify the procedure of knowledge generation, transformation, and development; (3) a context analysis to explicitly investigate the themes and develop tendencies of each transport mode within the topic of climate adaptation. This review focused on a systematic and comparative analysis of different transport modes along with a supply chain; hence, it exposed high association among the different modes in terms of climate change effect, adaptation measures, and threat analysis.

During the first phase, we confirmed the scope of this research as climate change, adaptation planning, and transport (Note: We recognised the possibility of other transport modes (e.g., oil pipelines are also regarded as a type of transportation), but we only focused on the main four modes (i.e., sea, air, road, and rail)). Accordingly, the two groups of 10 keywords, included (1) "climate change", "impact", "risk", "analysis", and "adaptation" (Group 1); and (2) "transport", "seaport", "airport", "port", "road" and "rail" (Group 2). Then, those search strings and their substrings (e.g., transport, railway) were entered into the two searching platforms linked by an "AND" function, meaning the combinations of any keyword from each of the two groups. Subsequently, the search results (i.e., all the combination results) were initially analysed through an "OR" function, by which there were 1791 articles elaborated upon.

Afterwards, three filtering and refining stages were set for purifying the most relevant papers, as below. First, only the articles containing more than one keyword (at least one in Group 1 and one in Group 2) were regarded as the targets, assuring that the selected papers were relevant to climate change and transport. Second, only peer-reviewed journal articles were collected, ensuring this review's authority. Third, the remaining database was further assessed and screened by reviewing each paper's title, keywords, and abstract to filter out

unrelated papers to enhance the review's reliability. Finally, the three-stage selection procedure helped yield 160 highly relevant representative articles published since 2000.

Before the second phase, the selected papers were categorised into four groups regarding transport modes (sea, air, road, and rail). Furthermore, publication and authorship analysis were conducted regarding the publication numbers of each year, citation trends, dominant journals, geographical location, and collaboration of researchers. In particular, the circumstances of each transport mode were specifically evaluated to demonstrate the similarities and differences. Another novelty of this study is that we employed a context analysis by examining the details of these studies regarding related transport modes. By doing so, the evolving pattern of the different transport systems during the research period was revealed together with existing achievements and research gaps to facilitate future methodological innovation exploration in Section 3. To minimise omitting essential knowledge, this paper employed an author-based citation analysis approach [43] to supplement the above purified database in the last phase. This meant that after the current database development, the top 10 leading authors were analysed as the search to find their publication records and further collect relevant papers to ensure the inclusion of all key articles as comprehensively as possible. The method used to identify the leading authors referred to the co-authorship analysis across climate change research and adaptation for different transport modes [39,40].

### 2.2. Critical Review and Empirical Results

To interpret different dimensions of transport research on climate adaptation, publication tendencies were first discussed, including publication numbers, citations, dominant journals, and the geographical locations of authorship. Next, a specific assessment was undertaken to reveal each transport mode's research focus and characteristics.

### 2.2.1. Publication Trends

By counting the publication year of all related publications, the analysis in this part provides readers with an overview of the evolving themes and patterns, particularly the research tendency of each transport mode, with the latest information on this subject.

Through the critical analysis of the selected papers, in general, there were two periods divided by 2013, showing a significant pattern difference. As shown in Figure 1, in the first phase (i.e., from January 2006 to December 2012), the total number of publications was only 11. However, since 2013, there has been a noticeable rise, while the number has started to fluctuate, growing from seven to 29. This was possibly because of the impact of COVID-19, as much transport research efforts (e.g., research grants and special issues leading transport journals) in terms of dealing with the emerging challenges presented by the pandemic over the past 2–3 years. Maintaining a high publication rate of 13.5 per year, the second phase (from January 2013 to April 2023) contributed 93.1% of the total papers, with a peak in 2020 when 29 papers were published. The fast-growing pattern over the recent ten years indicates the strong developing potential of this research area.

Among the papers, sea and road transport were the domain segments, in which 89 studies related to climate risks and adaptation in these two transport segments were found, occupying 56% of the total. However, as elaborated upon in Figure 2, less attention was given to air and rail transport, with only seven and ten papers being presented, respectively. It was worth noting that relatively large proportional studies involved more than one mode. That being said, 28% of the publications stemmed from the multi-modal industry.

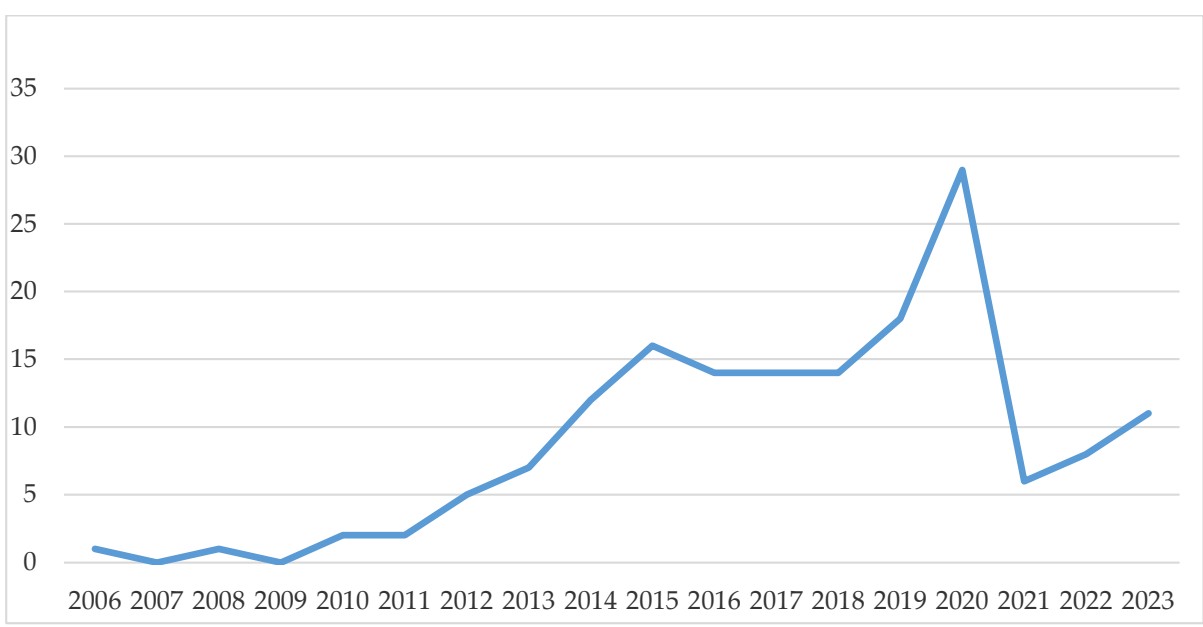

**Figure 1.** Number of papers by year of publication (January 2006–April 2023).

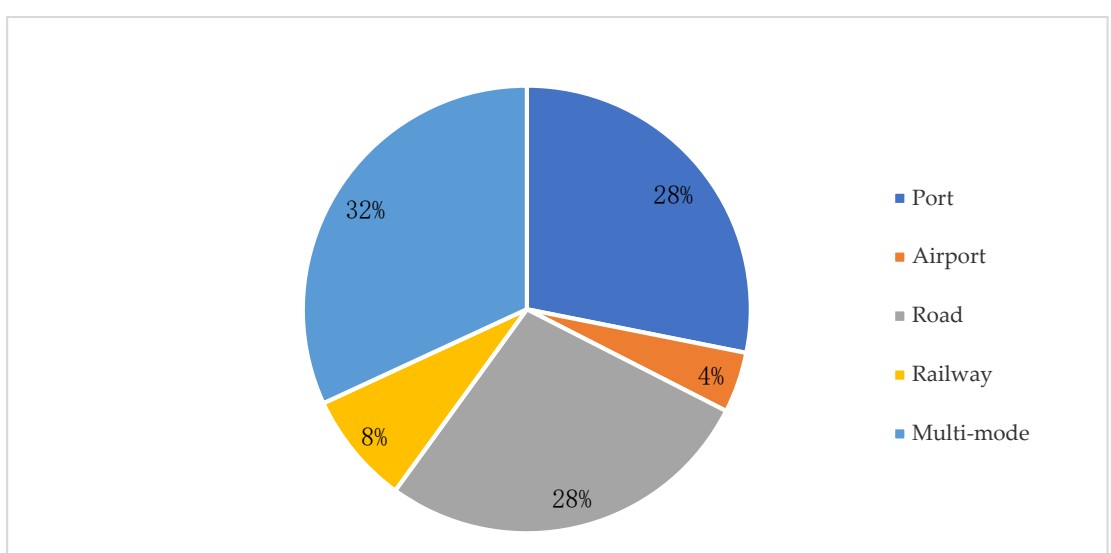

**Figure 2.** Percentage of papers by transport mode (January 2006–April 2023).

The citation trend is another index that allows the measurement of the contribution of climate adaptation research in transport-related fields. We found that the citation number was highly associated with the distribution of the publication of each mode. The top citation was for the multi-modal sector, with 586 citations, beginning from 35 records in 2006, followed by 375 for sea, 250 for road, 108 for rail and 18 for air transport. Thus, it was suggested that multi-modal research attracted attention, but systematic review and analysis were relatively scarce. Meanwhile, there was a relatively mature research pattern in terms of sea and road transport compared with rail and air transport. The latter, especially air transport, had developed slowly and might be overlooked by researchers, industrial professionals, and the government. This is possibly because the current transport adaptation focuses more on infrastructures than operations. Road and railway are associated with many studies, while seaports are often treated as the node and gateway of large door-to-door supply chain services, closely linked with road and railway. Relatively speaking, scholars are more concerned about airport operations than infrastructure, which

partially justifies why more attention is given to infrastructure than operations in terms of existing transport adaptation.

This study further traced the source of journals that published the most papers during the study period. Table 1 lists the top 12 journals, where the *Transportation Research Record* and *Transportation Research Part D: Transport and Environment* were the pivotal contributors with ten papers each, followed by the *Journal of Infrastructure Systems and Climatic Change*. Simultaneously, other journals as listed, such as *Sustainability*, *Transport Policy*, *Maritime Policy and Management*, and *Regional Environmental Change*, generated more than three articles each. The 12 journals contributed a total of 71 papers which made up nearly half of the total number. Thus, the multifaceted source of dominant journals reflected the variety of subjects in this research topic, including but not limited to transport and infrastructure, climate change, sustainability, policymaking, geography, and engineering.

**Table 1.** Dominant journals of climate adaptation research in the transport sector (January 2006–April 2023).

| Journal Name | Scopus 2-Year Impact Score | Number |
|:---:|:---:|:---:|
| *Transportation Research Record* | 2.06 | 10 |
| *Transportation Research Part D: Transport and Environment* | 7.24 | 10 |
| *Journal of Infrastructure Systems* | 3.71 | 8 |
| *Climatic Change* | 4.59 | 8 |
| *Sustainability* | 4.17 | 6 |
| *Transport Policy* | 6.36 | 6 |
| *Maritime Policy and Management* | 3.41 | 5 |
| *Regional Environmental Change* | 4.40 | 5 |
| *European Journal of Transport and Infrastructure Research* | 1.19 | 4 |
| *Journal of Transport Geography* | 6.02 | 3 |
| *Natural Hazards* | 3.14 | 3 |
| *Proceedings of the Institution of Civil Engineers: Civil Engineering* | 0.66 | 3 |

To identify the geographical distribution of climate adaptation research in transport, we recognised each paper's first and/or corresponding author as who was usually regarded as the primary author(s). By doing so, we could understand the popularity and accessibility of this research topic in different regions. As is demonstrated in Figures 3 and 4, the corresponding researchers primarily came from more than 20 countries. However, researchers from the UK, the US, and Canada were the main forces, with 59 papers published over the past 16 years.

Specifically, researchers from the US and Canada formed a pivotal team focusing on the climate adaptation of sea and road transport since 2008. Meanwhile, research on climate adaptation in the rail sector has been predominantly carried out by scholars from Europe, especially the UK, since 2014. The background of the researchers was diversified in geography, including European (e.g., the UK, Germany, the Netherlands), North American (i.e., the US and Canada), and Australian backgrounds. As for multi-modal transport, American and British scholars still played the dominant role, while researchers were less regionalised, scattered from other European countries (e.g., Sweden), and from Asia (e.g., China) to Africa (e.g., South Africa). Thus, it is evident that developed countries (e.g., the US, UK, Canada) have conducted considerable research on the topic in the sea, road, and rail transport sectors. There is enormous potential for climate adaptation research on air transport and shipping to develop research opportunities for international collaboration in multi-modal fields.

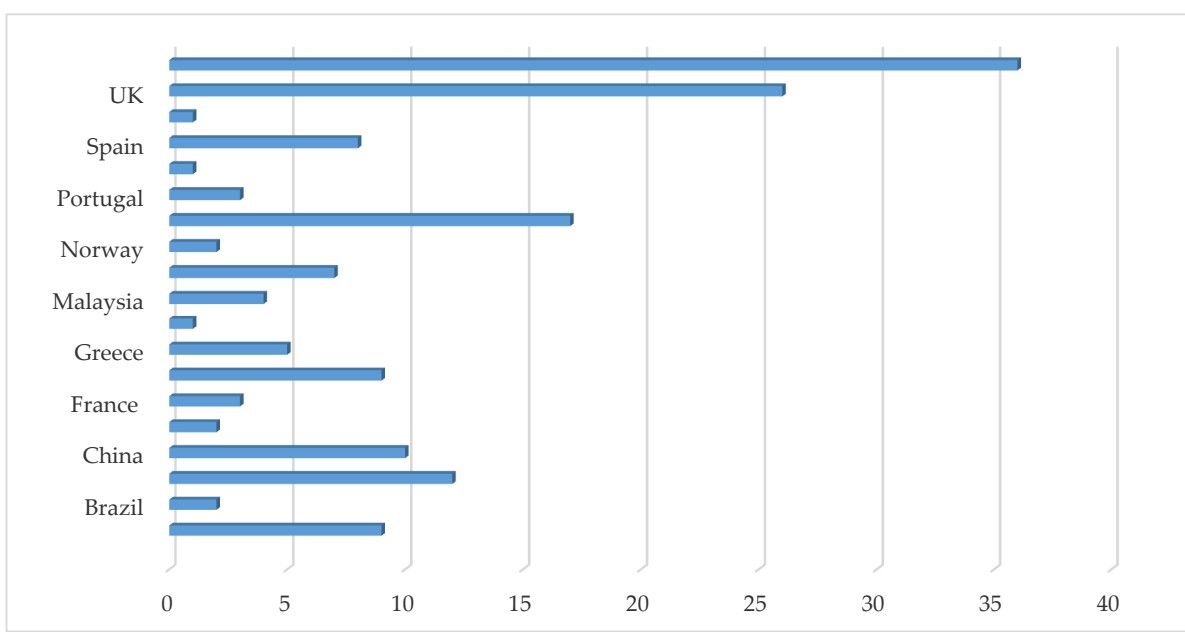

**Figure 3.** Number of articles by geographic location of the corresponding author (January 2006–April 2023).

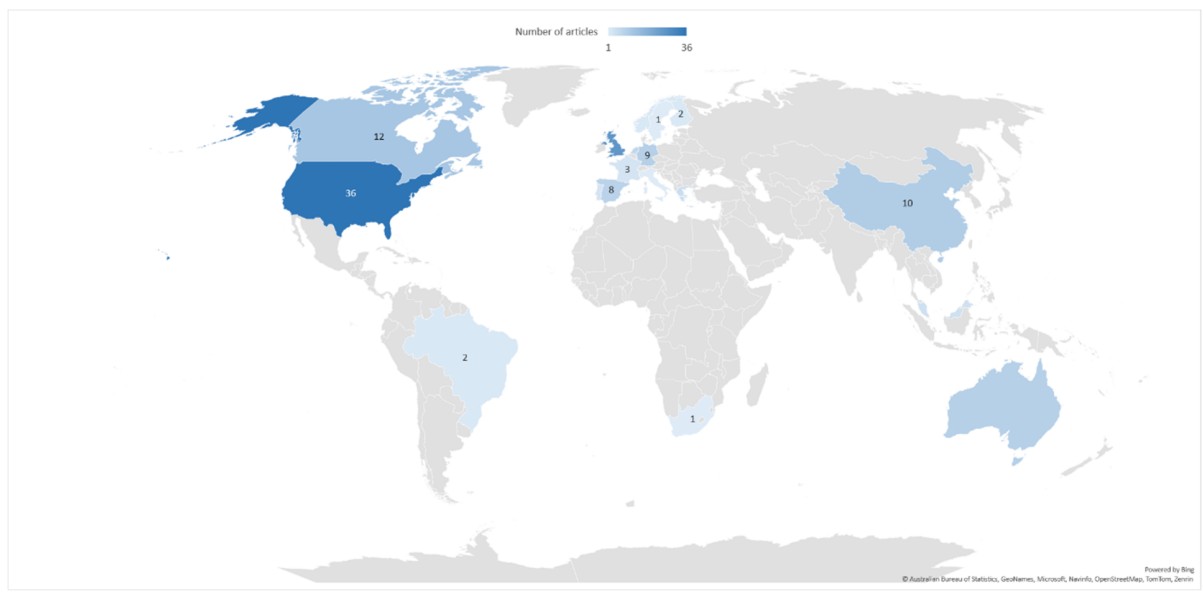

**Figure 4.** Map graph of number of articles by geographic location of the corresponding author (January 2006–April 2023).

### 2.2.2. Research Focused on Each Transport Mode

Acknowledging the overall trend in climate adaptation research in transport, one can categorise the selected articles into five groups based on their corresponding transport mode: sea, air, road, rail, and multi-modal. The evolving pattern, current situation, and dilemmas of each transport sector were investigated through context analysis to provide meaningful insights for future research on climate adaptation in transport.

#### Sea Transport

The first article linking climate change and seaports can be traced back to 2010. Focusing on the impacts of climate change on maritime navigation, Hawkes et al. [44] discussed how the changes in precipitation, sea level rise, and velocity could influence vessels, ac-

cessibility, transport infrastructure, and operation both positively and negatively. Also, they mentioned the utilisation of climate adaptation measures to respond to these impacts, which were clustered into diverse types of adaptation strategies and applied to the seaport sector in northwest Germany by Osthorst and Mänz [45].

In 2013, Becker et al. [8] published a paper on the impacts of climate change and adaptation planning in seaports, entitled "*A note on climate change adaptation for seaports: a challenge for global ports, a challenge for global society*". At the time this study took place, it was the highest cited article, stressing for the first time the strategic role of seaports and their vulnerability to climate risks (e.g., sea level rise (SLR) and storms) [8]. Furthermore, the results from collecting the opinions of worldwide seaport stakeholders concerning climate change impacts and adaptations suggested it was emergent to adapt these impacts through "soft" and "hard" measures with global efforts.

Furthermore, Becker et al. [4,9] explored how seaport resilience could be strengthened to minimise climate risks by requiring adaptation strategies from associated experts within two US seaports (Port of Providence and Gulfport). They suggested a seaport cluster composed of all of the relevant stakeholders (e.g., seaport authorities, governmental institutions, and insurance companies) to form a long-term master plan with solid leadership from "boundary organisations" [4,9]. Notwithstanding, until 2019, the stakeholders in the case of Providence still held vague perceptions about who should take a leadership role and offer significant investment for climate adaptation [6]. More recently, Mclean and Becker [11] have tended to determine the barriers challenging decision-makers to make resilience investments to climate and extreme weather hazards for seaports. Accordingly, the seven key adaptation barriers were typologies from extensive interviews among 15 seaports in the US (e.g., the absence of understanding of threats, funding, and communication).

Among the reviewed papers, although a considerable number of assessments have been made for measuring climate vulnerabilities to develop adaptation measures, most of these have applied qualitative techniques (e.g., interviews, focus groups, case studies) on small- or medium-sized scales. Some of these studies might be limited by their data representativeness and referential experiences for other regions beyond the investigated scope. In contrast, only a few articles, particularly after 2017, employed quantitative methods to evaluate climate risks and the effectiveness of adaptation measures, while they were coincidently highly cited. For instance, Yang et al. [38] developed a fuzzy Bayesian model to rank climate risks and the cost-effectiveness of adaptation measures with its applications in 14 container seaports in Greater China (Note: Greater China includes Mainland China, Hong Kong, Macau, and Taiwan). Meanwhile, there have been additional concerns about inter- and intra-port coopetition in terms of seaport adaptation investments, as well as the significance of the market structure of terminal operator companies in determining the size and timing of investment [46–48]. These studies modelled climate disasters under a general form of Knightian uncertainty, establishing a two-period options game model and conducting vulnerability analysis, especially in the condition of asymmetric information about actual climate disasters [49,50].

Overall, it is widely accepted by both academia and industrial practitioners that SLR, storms, and other climate-related hazards have been threatening diverse aspects of seaports (e.g., Panahi et al. [51]). Moreover, with growing evidence from scientific reports and research papers, adapting to climate change impacts has been put on the agenda for many seaport administrators worldwide, whilst many are stuck in its implementation [7]. Thus, the issues on how to stimulate the willingness to conduct adaptation planning and overcome dilemmas during the execution process have triggered considerable discussions in the recent decade. As the latest review paper by Loza and Veloso-Gomes [52] stated, how to integrate adaptation measures in the initial stage of new port design deserves to be considered as a larger project instead of focusing on narrow aspects or case studies. To tackle this outstanding challenge, topics such as employing quantitative risk assessment methods, building a standardised adaptation framework, and stimulating effective stakeholder collaboration are generating widespread concern associated with urgency.

Air Transport

The aviation sector has been considered to be a key contributor to climate change. The discussion regarding the relationship between climate change and aviation was initially revealed by Bows [53]. The article highlighted the significant role of the aviation industry in reducing greenhouse gas emissions based on a consensus projection that aviation would contribute up to 5% of global emissions by 2050. However, because airports are primarily situated in or close to urban areas, next to rivers, or alongside coasts, they are vulnerable to the impacts posed by climate change, such as flooding, SLR, increased temperatures, high winds, and extreme weather events [54,55].

Surprisingly, a few studies, mainly on a regional scale, recently started investigating how to respond to the impacts of climate change in airports with adaptation strategies. For instance, the authors in [54] illustrated the best practice, namely in Singapore Changi Airport, by implementing climate adaptation planning with solid governmental support. Through collecting data from 13 major Canadian airports, Zhao and Sushama [55] estimated the changes in temperature and wind influencing aeroplane take-off and landing performance. The study provided practical suggestions for flight operation with growing temperatures by examining three types of flights (long-, medium-, and short-haul) in diverse conditions (i.e., weight restriction days, strong tailwind, and crosswind). Another case study regarding climate risk assessment occurred at Athens International Airport in 2021 [56]. Combining historical climate data with adaptation data from surveys and interviews, they provided a list of risk evaluations with practical recommendations for outdoor workers, drainage systems, and infrastructure design to improve airport climate resilience.

A relatively comprehensive review of the studies regarding climate change impacts and adaption in the aviation sector was undertaken recently [57]. Analysing more than 40 relevant articles, the authors concluded that the emergent demand for adapting to the impacts posed by climate change had been inadequately researched nor considered by industrial practitioners in the air sector. The literature was considered highly relevant but multi-disciplinal (e.g., tourism is a commonly mixed discipline).

As stated by Tsalis, Botsaropoulou et al. [57] and Skouloudis, Evangelinos et al. [58], despite a rising number of sustainability reports being provided by airports, there were neither standard nor mandatory accounting principles for airports in reporting practices, which led to the failure of the assessment of disclosed information regarding climate risks. It partly explained the stagnancy of climate adaptation research in the aviation sector. Also, for an airport that has attempted to adapt to climate change impacts by enhancing the system resilience, measuring the resilience performance of airports in diverse extreme weather events is a challenge. It could be tackled by establishing a resilience metric measured by the speed of recovery, which was applied to a trial of the aviation system in China [59]. Echoed by Ryley, Baumeister et al. [60], research on climate adaptation in airports was generally constrained by complicated methods of crossing disciplines and the inaccessibility of researching the climate and airport data involved. Other gaps including the paucity of funding, qualitative investigations, and long-term timespan for adaptation planning were revealed. Therefore, it is suggested to review the literature, better benchmark industrial standards, and offer practical recommendations through best practices for future climate adaptation planning in airports.

Road Transport

Behind sea transport, road transport was the second most investigated sector regarding climate adaptation. Through the analysis of the 44 selected papers, it was found that existing studies have mainly focused on the physical dimensions of transport infrastructure (e.g., road pavements, drainage, tunnels, and bridges) on a national or regional scale (e.g., Tighe et al. [61], Guest et al. [62]). Moreover, instead of conducting comprehensive literature reviews or using conceptual frameworks, researchers used quantitative methods,

mainly modelling, to deal with specific demands in transport networks due to the impacts posed by climate change.

One of the most popular articles was published in 2014, in which Schweikert et al. developed comprehensive software called the Infrastructure Planning Support System to assist in decision-making for long-term road infrastructure planning [12]. The system in Schweikert et al.'s work provided an effective tool for cost–benefit analysis by considering technical, economic, and social factors in both quantitative and qualitative ways, given its wide practical applications in more than 50 countries. Another recently published article assessed the impacts of increased groundwater levels on road pavements in coastal areas owing to SLR [63]. The study found that rising groundwater could flow into the unbound materials. Based on the multilayer elastic theory, a groundwater flow model was designed to assess pavement performance in diverse climate scenarios to determine the magnitude of fatigue and reduction in the rutting life of the pavement.

North America has been the hotspot region for road adaptation to climate change since 2008. The latest research topics have included an assessment of the construction of the Tibbitt to Contwoyto Winter Road in threshold freezing conditions due to permafrost peatlands in Canada [64], an investigation of a modelling approach for bridge deck design against a corrosion attack in major Canadian cities [62], as well as an examination of the potential economic effects through a climate adaptation measure (i.e., upgrading the asphalt binder) to improve pavement resilience against growing temperatures in Virginia, USA [65].

Different from sea and rail transport (to be further discussed in the next sub-section), climate adaptation research on road transportation involved a wider geographical distribution. In addition to central European (e.g., the Netherlands, Germany) regions, a few developing countries (e.g., Malaysia, Saudi Arabia) were included. For instance, Shahid and Minhans [66] conducted a literature review by connecting climate change with road accidents to examine how climatic factors could influence road safety in Malaysia. Another study related to road safety was undertaken in Saudi Arabia that quantified the costs of road traffic accidents because of increased death and injuries due to changing climate variables, namely, precipitation, temperature, and sandstorms [67]. The latest survey investigated the mobility impacts on the road posed by flooding and heavy precipitation within urban residences in Ghana, Africa [68].

Scholars have made some achievements over the past decade. Nevertheless, there are still gaps that are yet to be bridged in terms of climate adaptation in the road industry. Guest et al. [62] stated that profound discussions regarding road research are significant when transforming revealed technical issues into effective institutional policies and long-term adaptation plans. Meanwhile, with abundant mathematical models being established and applied to regional cases, it has been suggested to re-test the results of climate risk analysis in diverse regions and tailor utilisation in other and multi-modal transport systems [37].

Rail Transport

Papers concerning rail transport were much fewer than those regarding sea and road transport. This might be because the relevant literature had not been generated until 2014. In the meantime, the research had a significant geographical feature that was narrowed to European countries, while five out of the ten selected papers concerned British railways.

Only two articles in 2016 elaborated on the climate risk analysis procedure for the high-speed rail network in Malaysia and Singapore [69]. It critically reviewed the climatic variables impacting the Malaysian railway infrastructure. Increased temperatures, heavy rainfall and flooding, lightning, and high winds were considered threats causing the delay of rail services, deterioration in operation, and failures of asset systems.

An assessment regarding existing and future flooding impacts on European railway infrastructure was conducted by Bubeck et al. [70], who utilised an infrastructure-specific damage model to project annual flooding damage in each climatic scenario. They stated that over US$340 million in annual losses could be avoided by controlling global warming to 1.5 °C, confirming climate adaptation's significance for European railways. Other European

research included the investigation of critical barriers of organisations in transferring scientific climate change information into adaptation planning in the French rail system [71] and the evaluation of flooding risks on underground transport with an application in terms of the Barcelona metro lines through a hydrodynamic model [72].

A few studies have been undertaken concerning specific climate risks related to specific dimensions regarding the UK rail system. Jenkins et al. [73] studied how hotter weather could alter passengers' demand and service expectations on the London Underground. Their results showed that various adaptation measures of cooling down infrastructure temperatures were expected to reach the satisfied thermal conditions for most lines in the mid-21st century under a high-emission scenario. SLR was a significant climate hazard for the coastal Dawlish railway in the UK. Dawson et al. [74] investigated the correlations between rail incidents and SLR over the past century through a semi-empirical model, followed by investigation of the relationship between future projections. In southeast England, rail incidents triggered by higher temperatures and heatwaves were discovered, including but not limited to the sagging of overhead lines, the breakdown of electrical assets, and track buckling. By introducing failure harvesting, Ferranti et al. [75] found the different resilience of rail infrastructure systems to temperature over different seasons. Taking the Minnamurra Railway Bridge in Australia as a case study, Kaewunruen et al. [76] applied building information modelling to facilitate the resilience of the bridge in terms of climate adaptation regarding asset management, operation, and maintenance.

Meanwhile, bridge scours, as a primary risk threatening the rail network in the southwest of England and Wales. were modelled by simulating the causal chain between scour hazard and climate change [77]. More recently, Wang et al. [31] surveyed the critical climate risks in the UK rail network, using fuzzy Bayesian reasoning (FBR) for risk prioritisation. Damage of the bridge foundation and collapse were highlighted as the pivotal risks owing to flooding and landslips.

As the climate adaptation research in the rail system started relatively late, restricted to regional or national cases, its development is less mature, with a few gaps that have yet to be bridged. For instance, during climate risk evaluation in coastal lines, besides fiscal losses, indirect economic and other socio-economic costs that are hard to project precisely need to be factored into consideration [70]. Adaptation planning, in the meantime, demands cross-departmental involvement with a broad range of stakeholders [74]. Furthermore, given the deficiency of comprehensive climate adaptation studies on railways, some advanced risk assessment methods, such as the failure-harvesting approach [25] and FBR model [37], can be re-designed to fit the requirements of other rail projects and promote adaptation strategies.

## 3. The CCRI Assessment Framework on Multi-Modal Transport Systems

To integrate climate data into climate change risk assessments, a CCRI assessment framework has been developed by Poo et al. [35] within the context of seaports. In this section, the CCRI framework has been tailored, generalised, and extended from a specific seaport framework to a broader multi-modal transport scope. Moreover, it allows the assessment of other transport infrastructures, such as roads and railways, individually and/or collectively. The steps of the newly adapted CCRI framework include:

1. Defining the CCRI hierarchy by climate data and extreme event details by collecting gridded climate data within a specified region;
2. Setting the grades of each indicator by identifying a quartile of data in a specified region;
3. Implying the evidential reasoning (ER) approach for CCRIs of a particular region or transport mode;
4. Evaluating the climate risk of the region/transport mode using climate data against the lowest level indicators;
5. Assigning weights to the CCRIs in the hierarchy;
6. Synthesising the evaluation of each transport mode to a multi-modal transport level using ER in the entire investigated nations involving multiple regions.

By implementing the data from different organisations, including the Meteorological Office and the British Oceanographic Data Centre (see Supplementary Material File S1), this paper has presented the climate risks of different cities and transport infrastructures within a specified region (see Supplementary Material File S2 with London as an example). In this study, the UK was used as a case study for presenting the comparative analysis of climate risks in different transport modes. While the specific modelling development and expansion (e.g., data collection and categorisation) have been explained below, the methodological flow driven by ER for generating a final CCRI from multiple datasets can be found in Supplementary Material File S3. This study implemented the methodology for statistically comparing the climate risks of different transport modes. By applying the CCRI framework, it visualised the climate risks of different transport systems in various locations. From the study regarding climate extremes on European transport infrastructures [77], the analyses have been split into four infrastructure types: airports, seaports, railways, and roads. Forzieri et al. [78] have weighted different climate extremes for different infrastructure types. Therefore, the risks of transport infrastructures are different even if they are in the same location. The representative cities in the UK were chosen based on their size and location for assessment and comparison, including Birmingham, Cardiff, Edinburgh, Glasgow, Liverpool, London, and Manchester (Table 2). Two risky seasons, summer (June to August) and winter (December to February) [35], were chosen for analysing four transport modes first: sea, air, road, and rail. Also, multi-modal transport was included by synthesising the results by averaging the weights of climate extremes from four transport modes as a reference.

The CCRI for each transport mode in one investigated city/region had a CCRI score between 0 and 1. The UK climate data and the weights of climate extremes were used to develop the CCRI framework for the UK transport system. The existing climate datasets and future climate impact changes were collected from the Meteorological Office (Met Office) [79,80]. The upper part of Table 2 presents the result of recent climate conditions, which uses the existing climate datasets only. Then, the bottom part presents the future climate conditions by including the future climate impact changes based on the existing climate datasets.

Table 2 presents the CCRI assessment of the UK transport systems for different transport modes and seasons. The rankings of each city were given based on their vulnerability to climate change risks. The rankings were consistent across most cities and transport modes during winter, apart from a few exceptions. However, during summer, some cities experienced variations in their rankings across different transport modes, with Manchester being the most vulnerable in the airport mode.

The term "variation" in this context referred to the differences or changes in the rankings of different transport modes within a city or between different cities and seasons [36]. For instance, a city may have a different rank for its airport infrastructure compared to its railway infrastructure, indicating a variation in the rankings of the two transport modes. Therefore, variation was used as a reference for further investigation.

During winter, most cities had consistent ranking positions across different transport modes, apart from a few exceptions, such as Liverpool. However, Cardiff, Liverpool, London, and Manchester had rank variations among different transport modes during winter. Also, during summer, Cardiff, Liverpool, and London had rank variations among different transport modes, whereas Manchester experienced the most severe climate risks in terms of its airport infrastructure.

This finding provides a valuable snapshot of the climate risks faced by different transport modes in different cities, both currently and in the future. This information can be helpful for policymakers, infrastructure managers, and other stakeholders interested in developing strategies to improve climate adaptation and resilience in the UK transport sector.

**Table 2.** CCRI assessment of the UK transport systems under recent climate conditions.

| Season | City | Seaport | Rank | Airport | Rank | Road | Rank | Railway | Rank | Multi-Modal | Rank |
|---|---|---|---|---|---|---|---|---|---|---|---|
| **Recent climate conditions** | | | | | | | | | | | |
| **Summer** | Birmingham | 0.1784 | 4 | 0.1572 | 5 | 0.1927 | 4 | 0.2064 | 4 | 0.1837 | 4 |
| | Cardiff | 0.2831 | 2 | 0.2183 | 4 | 0.2577 | 3 | 0.2722 | 3 | 0.2582 | 3 |
| | Edinburgh | 0.1504 | 5 | 0.1333 | 6 | 0.1439 | 6 | 0.1495 | 6 | 0.1443 | 6 |
| | Glasgow | 0.133 | 7 | 0.1076 | 7 | 0.1282 | 7 | 0.136 | 7 | 0.1263 | 7 |
| | Liverpool | 0.2888 | 1 | 0.2272 | 3 | 0.2727 | 2 | 0.2891 | 2 | 0.2698 | 2 |
| | London | 0.2514 | 3 | 0.2388 | 2 | 0.2956 | 1 | 0.3157 | 1 | 0.2751 | 1 |
| | Manchester | 0.1387 | 6 | 0.5993 | 1 | 0.1849 | 5 | 0.1989 | 5 | 0.1741 | 5 |
| **Winter** | Birmingham | 0.1985 | 6 | 0.2237 | 5 | 0.1889 | 6 | 0.1738 | 6 | 0.196 | 6 |
| | Cardiff | 0.3971 | 1 | 0.3524 | 1 | 0.3369 | 1 | 0.3266 | 1 | 0.354 | 1 |
| | Edinburgh | 0.2455 | 3 | 0.2879 | 2 | 0.2349 | 3 | 0.2174 | 3 | 0.2457 | 3 |
| | Glasgow | 0.2219 | 4 | 0.2585 | 3 | 0.2098 | 4 | 0.1915 | 4 | 0.2195 | 4 |
| | Liverpool | 0.2824 | 2 | 0.2544 | 4 | 0.2369 | 2 | 0.2282 | 2 | 0.2507 | 2 |
| | London | 0.1995 | 5 | 0.219 | 6 | 0.195 | 5 | 0.1834 | 5 | 0.199 | 5 |
| | Manchester | 0.1835 | 7 | 0.204 | 7 | 0.1746 | 7 | 0.1599 | 7 | 0.1803 | 7 |
| **Future climate conditions** | | | | | | | | | | | |
| **Summer** | Birmingham | 0.3535 | 3 | 0.3424 | 3 | 0.4124 | 3 | 0.4269 | 3 | 0.4139 | 3 |
| | Cardiff | 0.4673 | 1 | 0.4062 | 1 | 0.4713 | 1 | 0.4832 | 1 | 0.4577 | 1 |
| | Edinburgh | 0.3412 | 4 | 0.31 | 5 | 0.3464 | 5 | 0.352 | 5 | 0.3377 | 5 |
| | Glasgow | 0.2624 | 7 | 0.2313 | 7 | 0.2664 | 7 | 0.269 | 7 | 0.2576 | 7 |
| | Liverpool | 0.4378 | 2 | 0.374 | 2 | 0.4331 | 2 | 0.4425 | 2 | 0.4226 | 2 |
| | London | 0.3308 | 5 | 0.3176 | 4 | 0.3807 | 4 | 0.3927 | 4 | 0.3555 | 4 |
| | Manchester | 0.2845 | 6 | 0.2678 | 6 | 0.3168 | 6 | 0.3243 | 6 | 0.2985 | 6 |
| **Winter** | Birmingham | 0.208 | 6 | 0.1982 | 6 | 0.1807 | 6 | 0.17 | 6 | 0.1879 | 6 |
| | Cardiff | 0.3522 | 2 | 0.2911 | 4 | 0.286 | 3 | 0.2796 | 2 | 0.3028 | 2 |
| | Edinburgh | 0.2784 | 5 | 0.2548 | 5 | 0.2432 | 5 | 0.2349 | 5 | 0.253 | 5 |
| | Glasgow | 0.3178 | 3 | 0.3231 | 2 | 0.2771 | 4 | 0.2623 | 4 | 0.2927 | 4 |
| | Liverpool | 0.3858 | 1 | 0.3459 | 1 | 0.3239 | 1 | 0.3117 | 1 | 0.3429 | 1 |
| | London | 0.1366 | 7 | 0.136 | 7 | 0.1247 | 7 | 0.1169 | 7 | 0.03167 | 7 |
| | Manchester | 0.3129 | 4 | 0.3095 | 3 | 0.2867 | 2 | 0.2774 | 3 | 0.2967 | 3 |

Comparing the recent CCRI indices with forecasting CCRI indices shows that climate risks are rising. Taking the average values in both timeframes as a reference, CCRI indices in summer are expected to rise between 54.97% and 134.68%. Also, the indices in winter have foreseen a change within −37.33% and 59.01%. Edinburgh, Glasgow, and Birmingham are expected to experience heavier and more frequent climate extremes in the future during summer, and Manchester is expected to experience a similar trend during winter even though the trend is expected to be statistically milder. Also, Cardiff and London, which are in the south, are expected to experience weaker and less frequent climate extremes in the future during winter.

## 4. Discussion on Multi-Modal Climate Adaptation Studies

This section outlines the discussion and implications based on the findings from the literature review in Section 2 and the methodological findings in Section 3. Discussing the results of the systematic review and the assessment framework together can facilitate collaboration between stakeholders in the transport sectors, including policymakers, researchers, and practitioners of various transport modes. This can help to foster a shared understanding of the challenges posed by climate change and promote collective action towards addressing them.

*4.1. Discussion of the Findings from the Systematic Review*

Based on the literature review in Section 2, this section compares the primary progress of diverse transport modes in climate adaptation, including the number of publications and citations, dominant journals (Note: the dominant journals here refer to the publication of more than three articles during the study period), and the main geographical location of authorship, focus, and gaps. Then, as shown in Table 3, it dissects the characteristics of each mode to boost future climate adaptation in terms of transport.

While each of the four studied transport modes faces unique challenges in adapting to climate change impacts, there are similarities and overlap in the climate risk issues they encounter. One of these issues is the uneven development of climate adaptation efforts across different transportation networks and geographical locations. For example, the research shows a relatively large body of literature on climate adaptation in terms of sea transport but less research on air and rail transport. Additionally, research on climate adaptation in transport tends to be concentrated in specific geographical locations, such as North America and Europe. This uneven development of research and adaptation efforts can create challenges for policymakers, practitioners, and researchers who try to develop effective and comprehensive strategies for climate adaptation in transport. Therefore, it is essential to recognise these differences and address them by conducting research and implementing adaptation measures across all transport modes and a wide range of geographical locations. It is particularly important when multi-modal transport for container supply chains plays an increasingly crucial role in realising intercontinental door-to-door logistics services due to the development of international e-commerce.

Moreover, the countries pioneering adaptation in one transport mode are not necessary in an advanced position in the other modes, and it is crucial to develop rational adaptation measures by taking the advantages of different modes in tackling different types of extreme climate risks. For example, the UK has conducted empirical adaptation measures to respond to seaports' climate risks, and those studies are rarely considered as connected transport facilities and channels. However, in most circumstances, the threats of climate change relate not only to a single transport mode but also multiple networks within the area. The phenomenon triggers the demand for "alternative routing" for climate risks. For instance, in the UK, to track the impacts posed by a critical storm in Devon's rail system in the winter of 2014, National Express added 'rail replacement' coach services, and Flybe provided three extra flights from Newquay to Gatwick per day [81].

Understanding such, scholars are exploring the possibilities of enhancing climate resilience for different transport modes and regions. For example, Trinks et al. [82] mentioned that the road system is mainly affected by ice and snow, while strong winds and winter conditions mainly disrupt the rail system. Doll et al. [83] started considering diverse transport modes by identifying corresponding hazards and possible adaptation measures. There have been similar studies in the UK [84], Sweden [85], Italy [86], Greece [87], Tanzania [88], Austria [89], and the US [90]. Also, there have been comparative analyses among different countries, and there is a European climate risk project to project the risks of different transport infrastructures [91].

Still, there is a significant gap between quantifying the climate risks and analysing multi-modal systems, such as container supply chain (CSC) [92] and mobility as a service (MaaS) [93], on the same platform. A previous review of the road and rail industry [39] illustrated that several quantitative models have been adopted in multiple transport networks. However, many have been neither designed explicitly for the transport industry nor could they create a standardised solution to climate adaptation planning. Moreover, due to the high uncertainty of climate change, existing research has been restricted by the near-sighted planning timespan and hard-to-predict climate risks for future adaptation planning in the long term.

**Table 3.** Primary progress of diverse transport modes in terms of climate adaptation.

| Mode | Publication No. | Citation No. | Dominant Journals | Main Geographic Location of Authorship | Main Features | Gaps |
|---|---|---|---|---|---|---|
| Sea | 44 | 374 | *Maritime PolicyandManagement* | North America | Adapting to climate change impacts has been put on the agenda for many seaports; there have been a few assessments measuring climate vulnerabilities and adaptation measures; most have been qualitative methods. | Large-scale case studies; quantitative risk assessment method; implementation of adaptation measures; standardised adaptation framework; effective stakeholder collaboration. |
| Air | 7 | 18 | N/A * | Global | A few regional studies have recently started investigating climate change impacts in airports with adaptation strategies. | Complicated methods; sufficient funding; qualitative investigations; long-term timespan of adaptation planning. |
| Road | 44 | 250 | *European Journal of Transport and Infrastructure Research*; *Transportation Research Record* | North America | Mainly focuses on physical dimensions of transport infrastructure on a national or regional scale; most have been quantitative methods. | The transformation from technical issues into effective institutional policies and long-term adaptation; result testing of climate risk analysis in diverse regions and other transport systems. |
| Rail | 13 | 111 | *Climatic Change* | Europe | Research starts relatively late; significant geographic features (European countries). | Complex projection of indirect economic and other socio-economic costs; cross-department involvement with a broad range of stakeholders; advanced risk assessment methods. |

There was no dominant journal (more than one paper published) due to the limited number of articles.

*4.2. Discussion and Implications of the Findings from the CCRI Framework*

The UK case study in Section 3 provides a new methodological solution for evaluating the climate change risks of the multi-modal transport system for different cities. In the future, related studies can be extended to multi-modal systems by integrating network analysis with such assessments. Also, it is possible to implement CCRI indices in a global shipping network with different cities as nodes. Therefore, climate change impacts on multi-regional systems can be assessed.

Apart from assessing transport systems, governments and practitioners should work together on adaptation plans consisting of different transport modes. For example, the road [94] and rail networks [95] can diversify some container flows for container shipping networks. Indeed, climate mitigation and adaptation strategies have been simultaneously adopted in many cases, since the costs of deploying transport infrastructure and measures of realising carbon unlocking are highly relevant to the philosophy of reducing GHG emissions, carbon pricing, and long-term climate adaptation planning [96,97]. Therefore, more effective and efficient adaptation measures could be obtained if different transport modes were integrated.

Similar new research tendencies can be drawn from a recent scientometric review on climate change and the transport domain through knowledge mapping showing that resilience and sustainability have been researching hotspots with new technologies and best practices involved. Although different levels of capacity and experience on transport adaptation measures and planning still exist across different countries, the gaps are becoming smaller, given that the countries with a backseat role are catching up by taking excellent opportunities to learn best practice from those advanced and pioneering nations in this field to contribute wisdom, particularly for long-term planning in climate adaptation on transport.

## 5. Conclusions

Climate change adaptation has gained attention from researchers and practitioners but has yet to reach the level of intergovernmental and interdisciplinary cooperation. This is different to the climate change mitigation that has been achieved. Despite the widespread goal of carbon neutrality, the severity of climate threats has been consistently increasing since climate change adaptation was first introduced a decade ago. It therefore requires a larger scale and faster pace to stimulate transport adaptation research to ensure that climate risks are manageable in the coming decades. Therefore, this paper provides a comprehensive review of climate adaptation research in the transport sector over the past two decades, covering the latest studies across all transport modes. Recognising the uneven development of different transport systems, it pioneered a new CCRI framework to propose the integrated analysis of multi-modal transport systems across different regions. It has provided useful insights for future climate adaptation in the air and rail transport sectors with reference to the relevant experience gained from road and sea transport areas. The main challenges within these sectors are the use of complex approaches and inaccessible data, as well as the involvement of multiple disciplines, insufficient cross-departmental coordination, and a lack of financial support. In contrast, the sea and road transport industries have seen significant development, but concerns regarding transforming the technical issues identified from climate risk analysis into effective policymaking and long-term planning require further investigation.

Also, the study investigated the interdependence of climate risks on diverse transport modes, focusing on multi-modal transport networks. This approach acknowledges that minor impacts caused by climate change at a single location could significantly influence the entire transport system and its surrounding areas due to cascading effects that inherently characterise these types of climate risks. Given the existing fragmented and unbalanced research on transport adaptation to climate change, this study sought to provide a novel perspective by linking all primary transport modes within a holistic climate adaptation assessment framework. Within this context, to support climate risk assessment as the

first and foremost phase in climate adaptation planning, the Climate Change Risk Indicator (CCRI) framework was constructed and applied to examine climate scenarios in the UK transport network. The novel framework provides researchers and practitioners with an overview of climate adaptation with the latest tendency, empirical insights, and a referable tool that enables decision-makers to utilise objective data to assist with quantitative risk analysis in rational transport adaptation planning.

Admittedly, this study might not have had access to all of the relevant data on climate adaptation in the transport sectors due to complex approaches and a lack of accessible data. Additionally, the review might not have fully addressed the transport sector's financial and political barriers to climate adaptation planning. Therefore, several possible future research directions can be put forward. First, more surveys and interviews should focus on climate change adaptation, including risk and cost assessments throughout all transportation modes. By doing so, the cross-fertilisation concept could be extended to climate change adaptation measures for different transport modes and facilitate a better understanding of each mode's unique challenges and opportunities. Second, the network analysis could be introduced to optimise climate resilience nationally, regionally, or even globally due to the limited finances of different countries for climate risk management. This direction can provide policy recommendations to countries and global authorities, such as the United Nations and the World Bank, to promote collaboration and cooperation across borders. Finally, examining case studies of climate change adaptation initiatives can provide insights into the political barriers faced by policymakers, practitioners, and other stakeholders. This direction can identify the factors contributing to success or failure in implementing climate adaptation policies and programs, and highlight best practice and lessons learned.

Thus, this paper presents a call for multi-modal cooperation for evaluating the climate resilience of the transport systems. The literature review and the CCRI framework provide the needs and possibilities for scholars, policymakers, and other professionals to investigate the transport system in diverse ways. Transport is essential for maintaining human life and well-being. Thus, it is necessary to adapt to the changing climate faster than before. This paper provides a useful platform to integrate the climate risk evaluation of different transport modes in different regions and calls for the relevant contributions for realising the overall multi-modal transport system's adaptation to climate change from a global perspective.

**Supplementary Materials:** The following supporting information can be downloaded at: https://www.mdpi.com/article/10.3390/su15108190/s1. See Files S1–S3 [98–104].

**Author Contributions:** Conceptualisation, T.W. and M.C.-P.P.; Methodology, M.C.-P.P.; Software, M.C.-P.P.; Validation, M.C.-P.P., Z.Y. and A.K.Y.N.; Formal analysis, M.C.-P.P.; Investigation, M.C.-P.P.; Data curation, T.W.; Writing—original draft, T.W.; Writing—review and editing, T.W., Z.Y. and A.K.Y.N.; Visualization, M.C.-P.P.; Supervision, Z.Y. and A.K.Y.N.; Funding acquisition, T.W., Z.Y. and A.K.Y.N. All authors have read and agreed to the published version of the manuscript.

**Funding:** This research was financially sponsored by the Shanghai Pujiang Program (21PJC068) and the H2020 European Research Council [EU H2020 ERC-COG-2019 project (TRUST-864724)]. Adolf K.Y. Ng acknowledges the financial support received from the BNU-HKBU United International College (UIC) Research Funds (#R72021201).

**Institutional Review Board Statement:** Not applicable.

**Informed Consent Statement:** Not applicable.

**Data Availability Statement:** Data sharing not applicable.

**Conflicts of Interest:** The authors declare no conflict of interest.

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
