# Peer review of "Adapting to the Impacts Posed by Climate Change: Applying the Climate Change Risk Indicator (CCRI) Framework in a Multi-Modal Transport System"

_sustainability, doi:10.3390/su15108190_

Round 1
Reviewer 1 Report
The manuscript still has grammatical errors and typos.
It is necessary to
1) insert the bibliography in accordance with the journal template
2) better describe the novelty of the research in the introductory part
3) better describe the limitations and possible future research steps in the concluding part of the text
4) more commentary to accompany tables 2 and 3 is necessary
5) it is advisable to standardise the font of the texts of all images and figures
6) it is advisable to include some introductory concepts on sustainable and resilient cities by recalling for example the goals of Agenda 2030 or other global strategies .
7) We recommend reading the following research works to augment and improve the introductory part
1) van den Bergh, J. C. (2023). Contribution of Global Cities to Climate Change Mitigation Overrated. In The Barcelona School of Ecological Economics and Political Ecology: A Companion in Honour of Joan Martinez-Alier (pp. 335-346). Cham: Springer International Publishing.
2)Moraci, F., Errigo, M. F., Fazia, C., Campisi, T., & Castelli, F. (2020). Cities under pressure: Strategies and tools to face climate change and pandemic. Sustainability, 12(18), 7743.
3)de Abreu, V. H. S., Santos, A. S., & Monteiro, T. G. M. (2022). Climate Change Impacts on the Road Transport Infrastructure: A Systematic Review on Adaptation Measures. Sustainability, 14(14), 8864.
Author Response
Thank you very much for your valuable comments. Please find the corresponding yellowly-highlighted changes.
The manuscript still has grammatical errors and typos.
It is necessary to
1) insert the bibliography in accordance with the journal template
Thank you for the comment. The format is updated.
2) better describe the novelty of the research in the introductory part
Thank you for the comment. Please find the added words in Section 1.
“The novelty of this work lies in conducting a critical review of studies on how the transport sector adapts to the impacts of climate change across different transport modes and developing a Climate Change Risk Indicator (CCRI) framework for a comprehensive transport network. The framework aims to facilitate climate adaptation by comparing the climate resilience and vulnerability of different transport modes and enabling multi-modal adaptation planning. This approach provides a comparative analysis of the development of different transport modes in the climate change adaptation context, which can be useful for cross-sector fertilisation.”
3) better describe the limitations and possible future research steps in the concluding part of the text
Thank you for the comment. Please find the added words in Section 5.
“The study might not have access to all relevant data on climate adaptation in the airport and rail sectors due to complex approaches and a lack of accessible data. Additionally, the review might not fully address the transport sector’s financial and political barriers to climate adaptation planning. Therefore, several possible future research directions can be put forward. First, more surveys and interviews should focus on climate change adaptation, including risk and cost assessments throughout all transportation modes. By doing so, it can provide a chance to extend the cross-fertilisation concept to climate change adaptation measures for different transport modes and facilitate a better understanding of each mode’s unique challenges and opportunities. Second, the network analysis concept can be introduced to optimise climate resilience nationally, regionally, or even globally due to the limited finances from different countries on climate risk management. This direction can provide policy recommendations to countries and global authorities and promote collaboration and cooperation across borders. Finally, examining case studies of climate change adaptation initiatives can provide insights into the political barriers faced by policymakers, practitioners, and stakeholders. This direction can identify the factors contributing to success or failure in implementing climate adaptation policies and programs and highlight best practices and lessons learned.”
4) more commentary to accompany tables 2 and 3 is necessary
Thank you for the comment. Please find the added words for Table 2.
“The representative cities in the UK are chosen based on their sizes and locations for assessments and comparison, including Birmingham, Cardiff, Edinburgh, Glasgow, Liverpool, London, and Manchester (Table 2). Then, two risky seasons, summer (June to August) and winter (December to February) [35], are chosen for analysing four transport modes, seaports, airports, roads, and railways first. Then, multi-modal transport is also included by synthesising the results by averaging the weights of climate extremes from four transport modes as a reference.”
“Table 2 presents the CCRI assessment of the UK transport systems for different transport modes and seasons. The rankings of each city are given based on their vulnerability to climate change risks. The rankings are consistent across most cities and transport modes during winter, except for a few exceptions. However, during summer, some cities experience variations in their rankings across different transport modes, with Manchester being the most vulnerable in the airport mode.
The term “variation” in this context refers to the differences or changes in the rankings of different transport modes within a city or between different cities and seasons [36]. For instance, a city may have a different rank for its airport infrastructure compared to its railway infrastructure, indicating a variation in the rankings of the two transport modes. Therefore, variation is used as a reference for further investigation. During winter, most cities have consistent ranking positions across different transport modes except for a few exceptions, such as Liverpool. However, Cardiff, Liverpool, London, and Manchester have rank variations among different transport modes during winter. During summer, Cardiff, Liverpool, and London also have rank variations among different transport modes, whereas Manchester experiences the most severe climate risks in terms of its airport infrastructure.
The finding provides a valuable snapshot of the climate risks faced by different transport modes in different cities, both currently and in the future. This information can be helpful for policymakers, infrastructure managers, and other stakeholders interested in developing strategies to improve climate adaptation and resilience in the UK transport sector.”
Also, please find the added words for Table 3.
“While each of the four studied transport modes faces unique challenges in adapting to climate change impacts, there are also similarities and overlaps in the climate risk issues they encounter. One of these issues is the uneven development of climate adaptation efforts across different transportation networks and geographic locations. For example, the research shows a relatively large body of literature on climate adaptation in seaports but less research on airports and rail. Additionally, research on climate adaptation in transportation tends to be concentrated in specific geographical locations, such as North America and Europe. This uneven development of research and adaptation efforts can create challenges for policymakers, practitioners, and researchers who try to develop effective and comprehensive strategies for climate adaptation in transportation. Therefore, it is essential to recognise these differences and work to address them by conducting research and implementing adaptation measures in all transportation modes and a wide range of geographic locations.”
5) it is advisable to standardise the font of the texts of all images and figures
Thank you very much. All the texts in Tables and Figures are changed to Times New Roman.
6) it is advisable to include some introductory concepts on sustainable and resilient cities by recalling for example the goals of Agenda 2030 or other global strategies.
Thank you very much. Please find the added goal references in Section 1.
“Therefore, the goals of Agenda 2030, also known as the Sustainable Development Goals (SDGs), include a range of objectives related to climate change adaptation [3]. SDG 11, “Make cities and human settlements inclusive, safe, resilient, and sustainable”, is particularly relevant to climate change adaptation for transport infrastructure. This SDG includes the target to “strengthen efforts to protect and safeguard the world’s cultural and natural heritage”, which can be achieved through measures, such as improving the resilience of transport infrastructure to climate change impacts. Furthermore, SDG 13, which aims to “take urgent action to combat climate change and its impacts”, is relevant to climate change adaptation for transport infrastructure, as discussed earlier. This SDG includes the target to “integrate climate change measures into national policies, strategies, and planning”, which can help mainstream climate change adaptation considerations to transport infrastructure planning and design.”
7) We recommend reading the following research works to augment and improve the introductory part
1) van den Bergh, J. C. (2023). Contribution of Global Cities to Climate Change Mitigation Overrated. In The Barcelona School of Ecological Economics and Political Ecology: A Companion in Honour of Joan Martinez-Alier (pp. 335-346). Cham: Springer International Publishing.
2)Moraci, F., Errigo, M. F., Fazia, C., Campisi, T., & Castelli, F. (2020). Cities under pressure: Strategies and tools to face climate change and pandemic. Sustainability, 12(18), 7743.
3)de Abreu, V. H. S., Santos, A. S., & Monteiro, T. G. M. (2022). Climate Change Impacts on the Road Transport Infrastructure: A Systematic Review on Adaptation Measures. Sustainability, 14(14), 8864.
Thank you very much. They are all included in Section 1.
“In terms of mitigation, new energy alternatives, such as electric vehicles [16, 17] and ships [18, 19] and tactical management, such as speed control [20] and reverse logistics [21], are used to promote decarbonisation with diverse supports, including the application of urban policies [22] and circular economy concept [23].
In the post-pandemic era, the stresses between transportation and climate change have been tightening due to complex natural and human factors which increase the vulnerabilities of the urban system [24].”

Reviewer 2 Report
The manuscript examines how the transport sector can adapt to the impacts of climate change through a literature review and constructs a CCRI framework. The research process is reasonable and the findings have some practical value, but there are still several issues that must be addressed, as follows:
(1) It is already 2023, but the author's analysis of the literature only goes up to 2021, and the conclusions obtained may not be sufficient, so it is suggested that some relevant literature from the last 2 years be added.
(2) The authors seem to have neglected the impact of the infrastructure itself in their study of how the transport sector is responding to climate change[1,2], focusing more on the external risk factors and suggesting that this section be supplemented.
10.1016/j.trd.2016.11.002
10.3390/ijerph20021170
(3) The manuscript suggests that climate change refers to severe climate (if I understand correctly), but in the conclusion it mentions carbon neutrality, I don't see the relevance of this. Please add clarification to this question
Author Response
The manuscript examines how the transport sector can adapt to the impacts of climate change through a literature review and constructs a CCRI framework. The research process is reasonable and the findings have some practical value, but there are still several issues that must be addressed, as follows:
(1) It is already 2023, but the author's analysis of the literature only goes up to 2021, and the conclusions obtained may not be sufficient, so it is suggested that some relevant literature from the last 2 years be added.
Thank you very much. The literature review is updated to 2023.
(2) The authors seem to have neglected the impact of the infrastructure itself in their study of how the transport sector is responding to climate change[1,2], focusing more on the external risk factors and suggesting that this section be supplemented.
10.1016/j.trd.2016.11.002
10.3390/ijerph20021170
Thank you very much. They are added in Section 5.
“Indeed, it is noted that climate mitigation and adaptation strategies are simultaneously adopted in many cases, since the costs of deploying transport infrastructure and measures of realising carbon unlocking are highly relevant to the philosophy of reducing GHG emissions, carbon pricing and climate adaptation long-term planning [97, 98].”
(3) The manuscript suggests that climate change refers to severe climate (if I understand correctly), but in the conclusion it mentions carbon neutrality, I don't see the relevance of this. Please add clarification to this question.
Thank you. We have taken away the words related to carbon neutrality.
“Climate change adaptation has gained attention from researchers and practitioners, but it has yet reached the level of intergovernmental and interdisciplinary cooperation, different with that climate change mitigation has achieved. Despite the widespread goal of carbon neutrality, the severity of climate threats has been consistently increasing since climate change adaptation was first introduced a decade ago. It therefore requires a larger and faster pace to stimulate transport adaptation research to ensure the climate risks are kept at a manageable level in the next foreseen period. Therefore, this paper provides a comprehensive review of climate adaptation research in the transport sector over the past two decades, covering the latest studies across all transport modes. Recognising the uneven development of different transport systems, it pioneers a new CCRI framework to propose the integrated analysis of multi-modal transport systems across different regions. It provides useful insights for future climate adaptation in the airport and rail sectors with reference to the relevant experience gained from road and seaport areas. The main challenges within these sectors are the use of complex approaches and inaccessible data, as well as the involvement of multiple disciplines, insufficient cross-departmental coordination, and a lack of financial support. In contrast, the seaport and road industries have seen significant development, but concerns about transforming technical issues identified from climate risk analysis into effective policymaking and long-term planning require further investigation.”

Reviewer 3 Report
Dear authors,
This reviewer finds this manuscript interesting and well-written. Below you will find several observations of your paper. This reviewer considers that these observations will help you to generate a complete document. Moreover, please bear in mind that one of the purposes of a scientific paper is that the research carried out is replicable by other researchers. Please send back a new version of the paper and a response letter indicating how you attended each observation:
1. Please change the keywords that have already been written in the title, this to broaden the possibilities of searching and finding your paper.
2. Please, unify the font type in all the figures.
3. In Line 108, please check the writing of “investigates”.
4. In Line 111, it seems that the authors mean "modes" instead of "mode". Please check.
5. In Lines 116 and 117, the parentheses used in numerals 1) and 2) cause confusion with the parentheses of "(Group 1)" and "(Group 2)". This reviewer suggests using the numerals like 1. and 2.
6. In footnote 1 (page 3), it seems that the authors mean "main" instead of "man". Please check and correct.
7. In footnote 2 (page 3), please check the writing of “autorthors”.
8. In the legend of Figure 1, please consider changing the legend to "Figure 1. Number of papers by published year (January 2006 - June 2021)". This is because the authors are not presenting the "distribution" of a variable. This also applies to Figures 2 and 3.
9. In Figure 1, please move the label numbers up to make them legible.
10. In Line 174, please check the writing of “top citied”.
11. In Line 185, please check the writing of “justfy”.
12. In Line 186, please check the writing of “exiting”. It seems that the authors mean "existing".
13. In Line 284, please delete the double parentheses.
14. In Table 3, please check punctuation in the phrase: “Research starts relatively late; Significant geographic features (European countries)”.
15. For this reviewer, despite the explanation about the CCRI assessment framework by the ER approach, the processing that leads to the results presented in Table 2 is not clear. For this reason, this reviewer asks the authors to take at least one city as an example (it could be London as the authors include the information in Appendix 2) and present the detailed numerical calculation, to obtain the CCRI of Table 2 for all transport modes, including multi-modal. It would also be very useful to add a new figure with the methodology of that calculation. The authors mention that the methodological flow driven by Evidential reasoning ER for generating a final CCRI from multiple datasets can be found in Poo et al. (2021); however, this reviewer considers that it is not enough, and that this manuscript should present the complete methodology.
16. This reviewer asks the authors to include all the necessary explanations, using the requested example, so that other people can replicate their work and obtain the same results.
17. Please add a note to Appendices 1 and 2, clarifying what UB and LB are, and, in the explanation, clarify when one or the other should be used.
18. For Appendix 2, please specify in the manuscript what the authors mean by the term "linguistic assessment".
19. In Appendix 2, the authors present, for example, “2 (83%), 3 (17%)” for ID 3, or similar information for the other IDs. Please, when explaining the calculation example (observation 15) include what is the meaning of all the numerical information.
Author Response
This reviewer finds this manuscript interesting and well-written. Below you will find several observations of your paper. This reviewer considers that these observations will help you to generate a complete document. Moreover, please bear in mind that one of the purposes of a scientific paper is that the research carried out is replicable by other researchers. Please send back a new version of the paper and a response letter indicating how you attended each observation:
- Please change the keywords that have already been written in the title, to broaden the possibilities of searching and finding your paper.
Thank you very much for the advice. I have changed some keywords.
- Please, unify the font type in all the figures.
Thank you very much. All the texts in Tables and Figures are changed to Times New Roman.
- In Line 108, please check the writing of “investigates”.
Thank you very much. It has been checked.
- In Line 111, it seems that the authors mean "modes" instead of "mode". Please check.
Thank you very much. It has been checked.
- In Lines 116 and 117, the parentheses used in numerals 1) and 2) cause confusion with the parentheses of "(Group 1)" and "(Group 2)". This reviewer suggests using the numerals like 1. and 2.
Thank you very much. It is changed to use the numerals like 1. and 2.
- In footnote 1 (page 3), it seems that the authors mean "main" instead of "man". Please check and correct.
Thank you very much. It has been changed.
- In footnote 2 (page 3), please check the writing of “autorthors”.
Thank you very much. It has been changed.
- In the legend of Figure 1, please consider changing the legend to "Figure 1. Number of papers by published year (January 2006 - June 2021)". This is because the authors are not presenting the "distribution" of a variable. This also applies to Figures 2 and 3.
Thank you very much. It has been changed.
- In Figure 1, please move the label numbers up to make them legible.
Thank you very much. It has been changed.
- In Line 174, please check the writing of “top citied”.
Thank you very much. It has been changed.
- In Line 185, please check the writing of “justfy”.
Thank you very much. It has been changed.
- In Line 186, please check the writing of “exiting”. It seems that the authors mean "existing".
Thank you very much. It has been changed.
- In Line 284, please delete the double parentheses.
Thank you very much. It has been changed.
- In Table 3, please check punctuation in the phrase: “Research starts relatively late; Significant geographic features (European countries)”.
Thank you very much. It has been changed.
- For this reviewer, despite the explanation about the CCRI assessment framework by the ER approach, the processing that leads to the results presented in Table 2 is not clear. For this reason, this reviewer asks the authors to take at least one city as an example (it could be London as the authors include the information in Appendix 2) and present the detailed numerical calculation, to obtain the CCRI of Table 2 for all transport modes, including multi-modal. It would also be very useful to add a new figure with the methodology of that calculation. The authors mention that the methodological flow driven by Evidential reasoning ER for generating a final CCRI from multiple datasets can be found in Poo et al. (2021); however, this reviewer considers that it is not enough, and that this manuscript should present the complete methodology.
Thank you very much. Due to the length of assessment summary, I have attached the full methodology in Appendix 1.
- This reviewer asks the authors to include all the necessary explanations, using the requested example, so that other people can replicate their work and obtain the same results.
Thank you very much. Due to the length of assessment summary, I have attached the full methodology in Appendix 1.
- Please add a note to Appendices 1 and 2, clarifying what UB and LB are, and, in the explanation, clarify when one or the other should be used.
Thank you very much. It has been changed.
- For Appendix 2, please specify in the manuscript what the authors mean by the term "linguistic assessment".
Thank you very much., I have attached the full methodology in Appendix 1 for explanation. For "linguistic assessment" it means,
“All datasets are divided with respect to the five linguistic assessment grades {L1 “Low risk”, L2 “Moderately low risk”, L3 “Medium risk”, L4 “Moderately high risk”, L5 “High risk”} to facilitate the climate risk value evaluation in the ER algorithm”
- In Appendix 2, the authors present, for example, “2 (83%), 3 (17%)” for ID 3, or similar information for the other IDs. Please, when explaining the calculation example (observation 15) include what is the meaning of all the numerical information.
Thank you very much., I have attached the full methodology in Appendix 1

Reviewer 4 Report
This is an excellent work on the topic. Please follow the next recommendations:
1. Check out the redaction of footnotes 1 and 2. It is recommended to use technical language (for example, "scenarios" is more suitable than "theatres") and avoid the redaction in the first person (for example, in conclusion section: "We believe that this study...", change to "This study serves as the ideal platform...")
2. ¿It would be helpful to include the quality of the journals in Table 1? i.e., JCR or Scopus index
3. It would be helpful and more visually attractive if Figure 3 is shown as a map graph.
4. It would be helpful to general terms of the CCRI method were described in the text or added to Appendix 1 to make readers easily understand the results and where the numbers came from. I needed to look for the article Poo, M. C. P., Yang, Z., Dimitriu, D., Qu, Z., Jin, Z., & Feng, X. (2021). Climate change risk indicators (CCRI) for seaports in the United Kingdom. Ocean & Coastal Management, 205, 105580. to know the meaning of UB, LB, L1, L2, L3, L4, for example, or the meaning of 40th/60th, 30th/70th... Maybe a short version of the method. This is important because you stated in the conclusions: Transport, which is essential for maintaining human life and well-being, is necessary to adapt to the changing climate closer than before. Therefore, your work must be reproducible to extend the research in this area.
Author Response
This is an excellent work on the topic. Please follow the next recommendations:
- Check out the redaction of footnotes 1 and 2. It is recommended to use technical language (for example, "scenarios" is more suitable than "theatres") and avoid the redaction in the first person (for example, in conclusion section: "We believe that this study...", change to "This study serves as the ideal platform...")
Thank you very much. It has been changed.
- ¿It would be helpful to include the quality of the journals in Table 1? i.e., JCR or Scopus index
Thank you very much. Scopus 2-year impact score is added in Table 1.
- It would be helpful and more visually attractive if Figure 3 is shown as a map graph.
I have tried and realized it is not easy to present all the details due to the size of the countries are very different. I have added a map as Figure 4, and use Figure 3 together to provide and visualize details.
- It would be helpful to general terms of the CCRI method were described in the text or added to Appendix 1 to make readers easily understand the results and where the numbers came from. I needed to look for the article Poo, M. C. P., Yang, Z., Dimitriu, D., Qu, Z., Jin, Z., & Feng, X. (2021). Climate change risk indicators (CCRI) for seaports in the United Kingdom. Ocean & Coastal Management, 205, 105580. to know the meaning of UB, LB, L1, L2, L3, L4, for example, or the meaning of 40th/60th, 30th/70th... Maybe a short version of the method. This is important because you stated in the conclusions: Transport, which is essential for maintaining human life and well-being, is necessary to adapt to the changing climate closer than before. Therefore, your work must be reproducible to extend the research in this area.
Thank you very much. Due to the length of assessment summary, I have attached the full methodology in Appendix 1.

Round 2
Reviewer 1 Report
The manuscript still has some typos and grammatical errors once this is corrected the paper will be eligible for publication
Author Response
Thank you very much for your comments. It is double-checked, and typos and grammatical errors are revised.
Reviewer 2 Report
The author has responded well to the comments and I think it should be accepted.
Author Response

(The authors gave the same response as above.)

Reviewer 3 Report
Dear authors,
This reviewer finds this manuscript interesting and well-written, and the authors attended to most of the observations of the first review (particularly the most important observations). Some observations persist but they are easily solvable:
1. Please, unify the font type in all the figures.
2. Something happened to footnotes 2 and 3 because they are no longer in the manuscript. Please check and correct.
3. In Figure 1, please move the label numbers up to make them legible.
4. In the appendices there are some typos.
5. It seems that the titles of subsections 2.1.1 and 2.2.2 are not the correct titles.
6. In the References section there are instructions from the Journal Sustainability that should not be there.
Author Response

(The authors gave the same response as above.)
